# Immune Cells Are Differentially Modulated in the Heart and the Kidney during the Development of Cardiorenal Syndrome 3

**DOI:** 10.3390/cells12040605

**Published:** 2023-02-13

**Authors:** Imara Caridad Stable Vernier, Raquel Silva Neres-Santos, Vinicius Andrade-Oliveira, Marcela Sorelli Carneiro-Ramos

**Affiliations:** 1Laboratory of Cardiovascular Immunology, Center of Natural and Human Sciences (CCNH), Federal University of ABC, São Paulo 09210-580, Brazil; 2Bernardo’s Laboratory, Center of Natural and Human Sciences (CCNH), Federal University of ABC, São Paulo 09210-580, Brazil

**Keywords:** acute kidney injury, ischemia/reperfusion, immune system, cardiac alterations

## Abstract

Cardiorenal syndrome type 3 (CRS 3) occurs when there is an acute kidney injury (AKI) leading to the development of an acute cardiac injury. The immune system is involved in modulating the severity of kidney injury, and the role of immune system cells in the development of CRS 3 is not well established. The present work aims to characterize the macrophage and T and B lymphocyte populations in kidney and heart tissue after AKI induced by renal I/R. Thus, C57BL/6 mice were subjected to a renal I/R protocol by occlusion of the left renal pedicle (unilateral) for 60 min, followed by reperfusion for 3, 8 and 15 days. The immune cell populations of interest were identified using flow cytometry, and RT-qPCR was used to evaluate gene expression. As a result, a significant increase in TCD4+, TCD8+ lymphocytes and M1 macrophages to the renal tissue was observed, while B cells in the heart decreased. A renal tissue repair response characterized by Foxp3 activation predominated. However, a more inflammatory profile was shown in the heart tissue influenced by IL-17RA and IL-1β. In conclusion, the AKI generated by renal I/R was able to activate and recruit T and B lymphocytes and macrophages, as well as pro-inflammatory mediators to renal and cardiac tissue, showing the role of the immune system as a bridge between both organs in the context of CRS 3.

## 1. Introduction

Chronic diseases have emerged as major contributors to global mortality and morbidity. In this sense, cardiac arrhythmias are one of the most important remote complications following kidney disfunction [1]. Acute kidney injury (AKI) is mainly caused by ischemia/reperfusion (I/R), predisposing these patients to various dysfunctions including cardiac electrical disturbance [2]. The heart-kidney axis leads to a disease known as cardiorenal syndrome (CRS). CRS is a pathophysiological disorder that affects the kidney and heart, either acutely or chronically, triggering changes in one of the organs due to an injury to the other. Specifically, cardiorenal syndrome type 3 (CRS 3) is characterized by acute worsening of renal function leading to cardiac dysfunction [3]. 

AKI is an example of an inflammatory process where the innate and adaptive immune responses are involved [4]. These responses mediate the damage to the renal tubular cells and recovery of the renal alteration. Dendritic cells, monocytes/macrophages, neutrophils, TCD4+, TCD8+, and B lymphocytes are present in this renal injury [5]. The population and the activation status of immune cells change according to the degree of injury in the kidney [6]. On the other hand, it has been shown in an in vivo experimental model that left ventricular end-diastolic and end-systolic volumes increase 48 h after renal injury, in addition to an increase in relaxation time and decrease in shortening fraction. It has also been indicated that increased pro-inflammatory cytokine levels are associated with higher chronic cachexia, left ventricular remodeling, and mortality rates in patients with chronic heart failure [7]. 

Macrophages play a key role in the occurrence of kidney injury, as well as in tissue repair after damage [8]. Monocytes differentiate into macrophages upon recruitment to the injured kidney in M1 or M2 population. There is a greater likelihood of encountering a diverse spectrum of macrophages whose composition changes according to the tissue microenvironment [9]. Both TCD4+ and TCD8+ cells are involved in the development and maintenance of AKI. It has been reported that mice deficient in TCD4+ and TCD8+ cells showed attenuated renal ischemia/reperfusion (I/R), in addition to decreased adhesion of these cell populations to renal epithelial cells in vitro after hypoxia [10]. Renal injury by I/R promotes the participation of macrophages, neutrophils, and lymphocytes, particularly TCD4+ cells and B cells [11,12]. Activation of antigen-dependent T cells has been demonstrated in experimental models of renal I/R [13,14]. Also, CD4-deficient mice are more protected than CD8-deficient mice in models of AKI induced by I/R or cisplatin, with TCD4+ cells playing a key role in both scenarios [15]. In the context of renal I/R, divergent protection or aggravation results have also been shown using B-cell deficient mice [16,17]. Unlike T cells, which can promote or repair kidney damage, IgM-secreting B cells are known to be detrimental in this scenario [18]. Jang et al. reported an increase in B-cell traffic on the third day after injury to the damaged kidneys during the repair phase [17]. Evidence using mouse models suggests that IgM antibodies impair the reparative phase; nevertheless, if B cells are depleted, there is an improvement in the kidney repair process [19].

In the CRS setting, systemic inflammation favors recruiting different cells of the innate and adaptive immune system to participate in acute or chronic cardiac injury [20]. For example, in heart failure (HF) induced by myocardial infarction (MI), it is possible to observe an increase in interleukin 1 beta (IL-1β), tumor necrosis factor alpha (TNF-α), interleukin 6 (IL-6), and lectin 3 levels [21]. As part of the innate immune response, macrophages are indispensable for the formation of the cardiac lymphatic system as well as for proper electrical conduction [22]. Regarding T cells, several studies point out that they play an important role in HF setting [23,24]. In relation to the B cells, some studies have shown that rituximab or antibody-dependent B-cell deletion reversed myocardial hypertrophy and improved cardiac function using animal models of pressure overload via transverse aortic constriction or nonischemic cardiomyopathy, respectively [25,26].

Taken together, the present study hypothesizes that the acute inflammatory response induced by ischemic kidney injury activates macrophage and T and B lymphocyte populations, which in turn participate in alterations in the renal tissue, and subsequently in the cardiac tissue, as observed in CRS 3. Therefore, the aim is to characterize the immune cell populations mentioned above using the AKI model caused by I/R as an induction of a systemic inflammatory condition, and thus to analyze the impact of the innate and adaptive immune response on cardiac tissue. 

## 2. Material and Methods 

### 2.1. Animal Experimental I/R Procedure

Male C57BL/6J mice 6–8 weeks old (21–26 g) were provided from the Animal Facility of Federal University of ABC (São Paulo, Brazil) according to Ethical Committee Protocol CEUA/UFABC No.: 8644151220. The mice were kept in an artificial light/dark cycle of 12 h at a constant room temperature of 24 °C during the experiments, and with water and food supplements available at all times. The I/R protocol was conducted as previously described by Feitoza et al. [27]. Mice were first weighed and anesthetized with intraperitoneal injection (i.p.) xylazine hydrochloride and ketamine hydrochloride at 100 mg/kg and 200 mg/kg body weight, respectively, diluted in 0.9% saline solution. The renal pedicle was accessed by opening the abdominal cavity, the organs were then plastered in hydrophilic gas. After locating the left renal pedicle of the animal, all the adipose tissue was removed from the region, and the left pedicle was isolated with the help of tweezers. The renal pedicle was then occluded using a microvascular clamp (DL Micof, São Paulo). After placing the clamp in the renal pedicle, it was possible to observe the immediate change in color of the kidney due to the occlusion of local circulation, indicating the efficiency in inducing renal ischemia. The animals were kept in a thermal blanket for 60 min. After this time, the clamps were removed, and the viscera were repositioned in the abdominal cavity. The animals first had the peritoneum and then the skin sutured using 6–0 and 3–0 silk thread, respectively. The animals were kept warm by indirect lighting until their complete recovery from anesthesia. The mice were divided into four groups (n = 6 per group): sham (mice subjected to I/R surgical procedure except renal pedicle occlusion); I/R 3 days (mice subjected to I/R surgical procedure and reperfusion for 3 days); I/R 8 days (mice subjected to I/R surgical procedure and reperfusion for 8 days); and I/R 15 days (mice subjected to I/R surgical procedure and reperfusion for 15 days). After the surgical procedure, following the established reperfusion times, euthanasia was performed in these three groups, where, in each of them, one or two mice from the sham group was included as a control animal, respectively. The thoracic and abdominal cavities of the animals were opened and the left kidney, heart, renal lymph nodes, and spleen were removed for further analysis.

### 2.2. Isolation of Immune Cells and Flow Cytometry

#### 2.2.1. Tissue Digestion and Cell Preparation 

The left kidneys and hearts were perfused with PBS 1X to remove all of the tissue blood. Each sample was harvested at each time point, minced, and placed into wells of a 6-well plate containing 5 mL of ADS 1X + collagenase 1 mg/mL (Worthington Biochemical Corporation, Lakewood, NJ, USA). The tissues were digested by incubation in the shaker at 100 rpm for 30 min at 37 °C. The obtained suspensions were filtered using 70 µm cell strainers (BD Falcon, Corning, NY, USA) in DMEM medium + 10% FBS. All samples were pelleted using centrifugation with 37.5% Percoll at room temperature (20–25 °C) at 1500 rpm for 5 min and subsequently resuspended in FACS buffer (PBS 1X + 2% FBS). Cells were then centrifuged, and the pellets obtained were resuspended in red blood cell lysis buffer (NH4Cl 4.01 g; KHCO3 0.5 g; Na2EDTA 0.0019 g) and 5 mL of PBS 1X was added after 1 min of lysis. Following another centrifugation, the pellets were resuspended in PBS 1X for the next step of the protocol. 

#### 2.2.2. Flow Cytometric Analysis

Cell resuspensions were stained directly using fluorochrome-conjugated mouse-specific antibodies purchased from BioLegend (San Diego, CA, USA). All samples were stained with PECy7 conjugated anti-CD45 (clone: 30-F11, 1:400 dilution), TruStain FcX anti-mouse CD16/32 Ab (clone: 93, 1:100 dilution), and Zombie NIR Fixable Viability Kit (1:100 dilution), in order to examine all leukocytes, to block the FcR receptor, and to exclude dead cells, respectively. Samples were split in two, and half were used to stain for T and B cells, and the other half to stain for macrophages. The following antibodies were used to identify the B and T lymphocyte populations: APC conjugated anti-mouse CD19 Ab (clone 6D5, 1:200 dilution); PerCP conjugated anti-mouse CD4 Ab (clone RM4-5, 1:200 dilution); and PE conjugated anti-mouse CD8 Ab (clone 56–37, 1:300 dilution). The macrophage populations (M1 and M2 phenotype) were selected using the following antibodies: APC conjugated anti-mouse CD86 Ab (clone: GL-1); PE conjugated anti-mouse CD206 (clone: C068C2); PerCP conjugated anti-mouse F4/80 (clone: BM8); and FITC conjugated anti-mouse CD11b (clone: M1/70). The positive M1 and M2 macrophages population were determined after exclusion of doublet and dead cells and population gated on CD45 + CD11b + F4/80+ followed by fluorescence minus one (FMO) for CD86 and CD206, respectively. All antibodies to identify macrophages were used at a 1:100 dilution. The samples were incubated for 30 min at 4 °C protected from light. Flow cytometry was performed using FACS Canto II flow cytometer (BD Bioscience San Jose, CA, USA), and analyses were done using the FlowJo version 10.0.7. 

### 2.3. Gene Expression Analysis Using Quantitative Real-Time PCR (qRT-PCR) 

Total RNA extractions were performed from the left kidney and heart samples using Trizol (Invitrogen, Waltham, MA, USA) following the manufacturer’s instructions. The tissues were homogenized with Trizol and chloroform and lysed using PolyTron (Kinematica AG, Malters, Switzerland) until total degradation. Unlike the previous one, RNA extraction from the renal lymph node was performed by maceration of this tissue in PBS 1X, followed by centrifugation and resuspension in trizol and chloroform. All precipitated RNA were isolated from the aqueous phase and purified using centrifugation with isopropanol and ethanol 75%, consecutively. RNA quantification and integrity were assessed using NanoDrop Lite spectrophotometer (Thermo Fisher Scientific, Waltham, MA, USA) and agarose electrophoresis (18S and 28S bands integrity), respectively. For the reverse transcriptase reaction, 1 µg of total RNA (for the renal lymph node sample) or 2 µg of total RNA (for the kidney and heart sample) were applied for each sample, and the cDNA obtained as the final product was used to perform real-time PCR (Rotor-Gene Q, QIAGEN’s real-time PCR cycler, Qiagen, Hilden, Germany) to quantify gene expression. The levels of mRNA transcripts for cyclophilin A (housekeeping gene), interleukin 17 receptor A (IL-17RA), forkhead box P3 (Foxp3), T-box transcription factor 21 (Tbx21), IL-1β), CD38 and CD19 were analyzed through the primer sequences designed with Primer BLAST and BLAST NCBI (Table 1). 

### 2.4. Statistical Analysis 

All data were analyzed using the GraphPad Prism 6.0 software program (GraphPad Software Inc, San Diego, CA, USA) and were expressed as mean ± standard error of the mean (SEM). The one-way ANOVA test with Bonferroni correction for multiple post-hoc comparisons was used to compare the means of more than two experimental groups, and *p* < 0.05 was considered significant. 

## 3. Results

### 3.1. T and B Lymphocyte Populations Are Activated in Renal and Cardiac Tissue after Renal I/R 

The percentage of immune cells in our model were processed and analyzed using flow cytometry to determine any variation in the inflammatory response in the left kidney and heart. Given the low number of immune cells in both tissues studied, it was not possible to perform cell counts. Therefore, we sought an alternative strategy to ensure that the sample acquisition conditions were the same in all cases. To do so, we normalized all samples by acquiring the same number of events, thus maintaining the same standard in the experiments performed. The gating strategy used to identify the immune populations of T and B lymphocytes and M1 and M2 macrophages is shown in Figure 1A,B, respectively. The gate strategy used to exclude dead and doublets cells can be observed in the Appendix A. 

In relation to the kidney tissue, the percentage of live CD45+ cells (leukocyte marker) significantly increased after 3 days of reperfusion compared to the sham group (6.36 ± 1.50 vs. 1.17 ± 0.7, respectively) (Figure 2A). A decrease in the number of leukocytes of 45.96% and 24.20% was observed after 8 and 15 days of reperfusion, respectively, in relation to the I/R 3d group, caused by an attenuation of the local inflammatory process (Figure 2A). A significant increase regarding TCD4+ cells was noted 15 days after reperfusion by approximately 42% compared to sham (Figure 2B). TCD8+ cells showed similar behavior, with an increase in their numbers in the IR 15 d group compared to the other three experimental groups (Figure 2C). On the other hand, a decrease in the B lymphocyte percentage was observed over time in the groups in relation to the control group after 8 and 15 days of reperfusion by approximately 37.3 and 21.3%, respectively (Figure 2D).

The number of leukocytes in cardiac tissue after AKI showed a significant decrease after 15 days of reperfusion compared to sham mice (Figure 2E). A significant difference between the experimental groups evaluated was not observed in the case of CD45 + CD4+ and CD45 + CD8+ cells (Figure 2F,G). Nevertheless, a significant decrease of 70% was observed for CD45 + CD19+ cells (Figure 2H) at 8 days of reperfusion compared to the control group. So, it is possible to infer that reperfusion influences the number of leukocytes and different types of cells in kidney and cardiac tissue in mice, and that this effect changes over time and is associated with an “attenuation of the local inflammatory process”.

### 3.2. Renal Ischemic Injury Induces Macrophage Populations in the Renal and Cardiac Tissue

The diversity of macrophage functions has led to several classification systems. Two phenotypes are well defined, with lipopolysaccharides (LPS) or interferon gamma (IFN-γ) being able to activate macrophages via the classical pathway (M1 macrophages). Alternatively activated macrophages (M2 macrophages) are polarized by Th2 cytokines, interleukin-4 (IL-4), and interleukin-13 (IL-13) [9]. As a key component of the inflammatory response that determines tissue destruction or recovery, increasing evidence suggests that macrophages do not remain committed to a single activation state; they can return to a resting state that can subsequently be reactivated in another way [28]. 

Herein, CD86+ and CD206+ markers were used in the preselected CD11b+ F4/80+ populations (Figure 3) with the purpose of identifying the two main phenotypes of M1 and M2 macrophages, respectively. Figure 3A illustrates a significant decrease in M1 macrophages observed after 3 days of reperfusion compared to the other three experimental groups in renal tissue (29% in relation to sham group and approximately 31% compared to I/R 8 days and I/R 15 days). However, no significant difference was observed in the M2 phenotype among the groups studied (Figure 3B). No change was observed in M1 and M2 macrophage populations in cardiac tissue (Figure 3C,D).

Based on these results, we can indicate that local renal inflammation was able to modulate the pro-inflammatory macrophages, but the systemic inflammatory process did not guarantee that both macrophage phenotypes showed significant cellular expansion in the heart.

### 3.3. Gene Expression of IL-17RA, Foxp3, and Tbx21 Was Modified for Renal I/R in the Renal and Cardiac Tissue 

The gene expression of immunological markers was evaluated using qRT-PCR in order to complement the results obtained by flow cytometry. The gene expression of (IL-17RA) in the left kidney (Figure 4A) showed no differences between the groups evaluated. In contrast, I/R induced a significant increase in (Foxp3) and (Tbx21) after 15 days of reperfusion compared to sham (Figure 4B,C). 

The observed gene expression pattern of these markers in the cardiac tissue was different than in the kidney. In this case, ischemic injury was able to promote an increase of approximately 44% in mRNA levels for the IL-17RA after 8 days of reperfusion in relation to the control group (Figure 4D). Both Foxp3 and Tbx21 gene expression showed no significant differences in any of the experimental groups evaluated (Figure 4E,F). 

In this case, the data obtained reflects that an increased expression of Foxp3 and Tbx21 after 15 days of reperfusion in kidney tissue compared to the control group suggests that these markers may play a role in the inflammatory response to ischemic injury in kidney tissue. Whereas, in cardiac tissue, IL-17RA expression predominates, conferring an inflammatory immune response there.

### 3.4. Inflammatory Status Is Modulated in Both Heart and Kidney 

Because of the inflammation generated by AKI, there are dissimilar immune molecules that participate in this context. Therefore, the gene expression of other markers was evaluated. IL-1β is a pro-inflammatory cytokine induced and activated after tissue damage, and its activation results in the production of other cytokines, including TNF-α and chemokines, which together recruit immune cells to the injury site [29]. This made it interesting to study the surface protein, CD38, because it contributes to cell differentiation, cytokine release, migration, and apoptosis processes; thus, it is often used as a cell activation and differentiation marker [30,31]. In addition, the B cell marker, CD19, was studied due to the fact that these lymphocytes are involved in the inflammatory process.

As shown in Figure 5A, IL-1β mRNA levels increased at 8 and 15 days of reperfusion in the renal tissue, approximately 10- and 20-fold, respectively, compared to the sham group. The mRNA levels for CD38 showed similar behavior to that of IL-1β, where an increase in its gene expression was evidenced in the last two I/R groups studied compared to the control group (Figure 5B). However, CD19 mRNA levels showed a different profile, in which there was a significant decrease in its gene expression at 15 days of reperfusion by 16% in relation to the sham group (Figure 5C).

There was variation in the gene expression profile of the previously mentioned markers in cardiac tissue. As shown in Figure 5D, the mRNA levels of IL-1β increased significantly in the I/R 8 days group compared to the control group, thus corroborating previous data in another study [2]. There was no change observed between the groups in relation to CD38 expression (Figure 5E). These results indicate that there is a variation in the gene expression profile of these markers in renal and cardiac tissue, indicating a modulation of inflammatory response to ischemic injury.

### 3.5. Renal Ischemia/Reperfusion Was Able to Increase IL-1β Expression but Not CD38 in the Renal Lymph Node 

The gene expression of some immunological markers in the renal lymph node (RLN, a secondary lymphoid organ with respect to the kidney) was studied with the objective of verifying if there was variation in its immune response. In this case, mRNA levels of IL-1β showed a significant increase in its gene expression at 15 days of reperfusion (1.0 ± 0.51 vs. 8.78 ± 0.62, *p* < 0.05) (Figure 6A). However, CD38 did not show any modification of its mRNA levels in any of the experimental groups evaluated, as shown in Figure 6B. Therefore, these data suggest that I/R promotes the participation of several organs involved in the activation of the animal’s immune system, given the recruitment of pro-inflammatory mediators that initiate the immune response. 

## 4. Discussion

The present study has five main findings: (i) the inflammatory renal process generated an increase in TCD4+ and TCD8+ lymphocyte populations cells in the kidney, while only B cells contributed to the generated cardiac injury; (ii) a regulatory-type immune response predominated in renal tissue, although the inflammatory role had more of an impact in the heart; (iii) IL-1β acts as an inflammatory mediator in the renal tissue and ensures the recruitment of other immune cells involved in the context of CRS 3; (iv) the increase in the number of leukocytes in the 3 days of AKI is in concordance with similar results reported by other authors; and (v) B lymphocytes showed a different response to renal ischemic injury with a decrease in their numbers starting on the eighth day of reperfusion.

The observed increase in the number of leukocytes in the 3 days of AKI is in concordance with similar results reported by other authors. For instance, Miyazawa et al. demonstrated an infiltration of leukocytes into the occluded renal pedicle starting at 6 h of reperfusion and maintained until 72 h [32]. Also, Chen et al. showed that toll receptor type 4 (TLR4) activation can recruit a greater number of leukocytes to the ischemic kidney during the first 3 days of AKI induction [33]. Our results show how TCD4+ and TCD8+ cells increase during the repair phase of renal I/R. Several studies indicate that the different subtypes of TCD4+ cells [Th1, Th2, Th17 and Treg] start to increase during this reparative phase. Treg stands out, presenting greater infiltration in the kidney starting on the 10th day of injury, probably related to their protective function in the tissue restoration process [34,35]. Additionally, in an I/R model, Gandolfo et al. demonstrated that TCD4+ cells in general, and Treg in particular, show an increase in numbers in the ischemic kidney, and this same phenomenon occurs for TCD8+ cells [36], which is in accordance with our observations for both T cell populations. Another study realized by Tao et al., using an I/R unilateral model for 45 min, also observed an increase in Treg cells at 10 days after the induced surgical procedure [37].

The B lymphocytes showed a different response to renal ischemic injury with a decrease in their numbers starting on the eighth day of reperfusion. This result aligns with the study by Jang et al., in which the mice that underwent the I/R surgical procedure for 45 min showed a decrease in B cells starting on the 10th day, persisting until 28 days of reperfusion. They additionally obtained greater infiltration of T cells occupying the niche left by B lymphocytes, which is clearly evidenced in the previous result where an increase of both TCD4+ and TCD8+ is shown [17]. This phenomenon of balance between both groups of immune cells indicates that the B cells in our model migrate from the repair phase and have specific changes in the populations involved and activation status over time, thus allowing the T lymphocytes to infiltrate and expand more in the damaged kidney.

The cardiac tissue showed different inflammatory responses compared to the renal tissue, likely due to the fact that the injury in this study was focused on the kidney. The percentage of leukocytes recruited to the heart decreased during the healing phase, which contradicts the findings of Majid et al., who found an increase in these cells at 3 and 15 days of ischemia associated with heart failure [38]. This difference can be explained by the systemic inflammation caused by the renal injury impacting the cardiac tissue in the early stages but decreasing over time.

Both populations were unchanged with respect to CD45 + CD4+ and CD45 + CD8+ cells in cardiac tissue, perhaps because although T cells increase in models that directly cause cardiac injury such as coronary artery ligation and myocardial infarction, it has been reported that the increase in TCD4+ and TCD8+ lymphocytes in the heart results from a wide spectrum of possible causes depending on the context of activation and the timing of the responses [21,39]. Unlike T lymphocytes, the numbers of CD45 + CD19+ cells only decreased by the 8th day of reperfusion. In this regard, Adamo et al. indicates that the recruitment of B cells to the inflamed tissue is likely a result of specific modifications in the endothelium [40]. Furthermore, in using a murine model of permanent myocardial infarction or ligation of the descending left coronary artery, Yan et al. found that these cells reach a numerical peak between the 5th and 7th day of direct injury generated in the heart [41]. In contrast, we found a decrease in B cells in the heart in the I/R 8 days group in our study. This could be due to our model being different, with the injury being directly induced in the kidney.

The significant decrease in M1 macrophages at 72 h after injury in this work does not agree with that reported by Lee et al., in which it was evidenced that M1 infiltration increases from the onset of AKI and is maintained until the third day, and then decreases during the repair phase [42]. The result observed in this study may be due to existing left kidney atrophy and then an incomplete or defective adaptive repair attempt leading to the development of renal fibrosis and I/R progressing to possible chronic kidney disease (CKD). Another explanation could be that macrophages have phenotypic plasticity which changes according to the microenvironment generated by the inflammatory lesion, subsequent fibrosis, and repair phase [8,43]. 

Several studies demonstrate that TCD4+ cells constitute the main subset of T cells responsible for ischemia-induced AKI [10,34]. Th17 cell expression is known to depend on the transcription factors: signal transducer and activator of transcription 3 (STAT3) and retinoic acid receptor-related orphan nuclear receptor gamma (RORγt) [44]. In this regard, a study by Lee et al. using a 30 min I/R model and STAT3 knockout mice indicated that interleukin-17 (IL-17) levels decreased in this animal model 48 h after initiating injury, and that deletion of this transcription factor in Th17 cells increased IL-4 and IL-10 production as well as Foxp3 mRNA levels. With this result, the researchers suggested that the degree of inflammation due to I/R may be determined by a balance between Th17 cells and Tregs [45]. Based on this, it is possible that the non-modification of IL-17RA mRNA levels in our model is because the local renal inflammation generated a greater recruitment of Treg cells, and therefore the Th17 cells did not reach sufficient activation through STAT3 to have a significant change.

The observation was the same for the Foxp3 and Tbx21 transcription factors. This result can be due to two factors: (i) Foxp3 plays an important role during the repair phase protecting the kidney from inflammation [46]. This was evidenced in a study where subjecting the mice to a unilateral I/R for 45 min obtained an increase in trafficking Treg cells from 10 days of induced AKI. They also corroborated the beneficial role of Foxp3 in this stage due to a probable negative modulation mechanism of pro-inflammatory cytokines produced by other T lymphocyte subsets [36]. (ii) Tbt + Treg cells can be recruited to the inflammation site due to the action of Th1 cells which promote their expression of the CXCR3 chemokine receptor. Thus, it is evidenced how the induction of Tbx21 in response to IFN-γ can regulate Treg, justifying how two transcription factors with antagonistic activities show the same patterns in their genic expressions in this work [47]. Consequently, this increased number of Tregs has an impact on the Th17 cells, showing that there is compensation and regulation between the transcription factors involved in our model, leading to a mainly renal tissue repair response.

The increase on day 8 in reperfusion of IL-17RA mRNA levels in the cardiac tissue agrees with some studies which have shown that using MI models induces an increase in IL-17 gene expression from the first 24 h to 72 h [48], maintaining this behavior over long-term periods such as 14d after the generation of cardiac damage [49]. Although the experimental examples cited have the heart as the target organ of injury, differing from this work, they also observed a significant peak of IL-17RA mRNA in the I/R 8 days group. This may be because a role has been reported in the literature for this interleukin in the deterioration of cardiac tissue by directly contributing to the apoptosis of cardiac cells [50]. Furthermore, another possible justification for this result is the fact that there was no modification of IL-17RA gene expression in the kidney tissue influenced by a predominant increase in Treg cells, and in contrast, there is an increase in Th17 in the heart, but no change in the mRNA levels of Foxp3.

The levels of mRNA IL-1β increased at 8 and 15 days of reperfusion, which is consistent with the findings of Kim et al., who observed an increase in this interleukin’s gene expression from the 7th day in a glomerulonephritis model [51]. Alarcon et al. also reported an increase of this mRNA in the 8 days I/R group using the same experimental model as in this work [2]. Our results agree with their report and may be related to the secretion of this interleukin by M1 macrophages. However, in our model, there was a significant decrease in M1 macrophages at 72 h after AKI caused by an incomplete attempt at adaptive repair, and a “doubtful visual increase” in M1 numbers at 8 days and 15 days of reperfusion.

CD38 mRNA levels showed similar changes to those of IL-1β, where an increase in its gene expression was evidenced in the last two I/R groups studied compared to the control group. Shu et al. found that the induction of AKI by LPS administration in C57BL/6 mice caused an increase in CD38 mRNA levels one hour after treatment initiation, reaching the highest peak after six hours [52]. Another study by Ogura et al. evidenced the increased gene expression of CD38 in renal tubular cells using an experimental model of diabetic rats, indicating how mitochondrial oxidative stress related to the reduction of NAD+/NADH altered the expression of the gene in question, granting an important role to this protein in the development of diabetic kidney disease [53]. Although the reported study used different experimental models than the one applied in this work, both constitute examples of how CD38 is modulated by an inflammatory process. This immune marker is expressed in various immune cells, such as macrophages, dendritic cells, neutrophils, NK cells, and T and B lymphocytes, among others [54]. Based on the above, we can infer that the increase in CD38 mRNA in our study at 8 and 15 days of reperfusion is possibly caused by the predominant influence of the T cells being activated in the renal tissue, since T cells were also activated at these same ischemic times as shown in the flow cytometry results.

However, the CD19 mRNA levels showed a different profile, in which there is a unique modification of its gene expression at 15 days of reperfusion; this is in agreement with that obtained using flow cytometry for B cells, which also significantly decreased in this group compared to the control group.

Distinct from what was obtained in renal tissue, the IL-1β mRNA levels significantly increased in the I/R 8 days group in the heart. Alarcon et al. evidenced the same result obtained in this study, justifying this event with the fact that IL-1β plays an important role in developing cardiac diseases, since blocking the signaling of this interleukin once I/R is induced can prevent the appearance of cardiac alterations. Thus, its inflammatory action is corroborated as a link between poor renal tissue function and cardiac damage [2].

A different result was also observed for CD38 gene expression in cardiac tissue. In this case, Zhang et al. used a MI and ischemia/hypoxia model and observed an increase in the mRNA levels of the protein in question, indicating that CD38 can be modified in cardiac tissue under different ischemia/hypoxia conditions [55]. Although the literature shows variations of this gene expression in the distinct cell types of the heart (being high in endothelial cells, moderate in fibroblasts, and low in cardiac myocytes) using cardiac I/R as a model [56], no gene modification was seen in this work. A possible explanation could be that since the target organ is the kidney, the activation and modulation of CD38 in cardiac tissue will depend on systemic inflammation, which was not able to induce a strong enough immune response to trigger an increase or decrease in gene expression of this protein.

There are tissues which have a strategic location throughout the animal’s body as part of the immune system’s components, favoring a rapid and effective immune response, such as the RLN [57]. The RLN plays an important role during inflammation because there is leukocyte traffic towards it, thus promoting an adaptive immune response by presenting antigens to T cells and activating them. However, the main function of this organ in a healthy organism is directed toward maintaining tolerance against antigen diversity [58,59].

The IL-1β mRNA levels in this secondary lymphoid organ showed a significant increase at 15 days of reperfusion. Some studies indicate that IL-1β is induced by stimuli such as LPS, favoring its secretion and contributing to an inflammatory scenario. Both Duncan et al. and Doisne et al. observed how IL-1β increases its expression in lymph nodes as part of an inflammatory immune response to pathogen-associated molecular patterns (PAMPs) [60,61]. In the context of our study, IL-1β will respond to stimuli such as damage-associated molecular patterns (DAMPs) due to the generated I/R, which will trigger production of this interleukin in the RLN. The increase in this immune mediator, specifically in the last experimental group evaluated, may also be related to the Th1 cells which have pro-inflammatory action and increase at the same time in the kidney, evidencing regulation, certain dependence, and immunological interconnection between these molecules in our model.

This study had a few limitations. (1) The sample size of immune cells obtained from the heart and kidney was limited, which could be improved by using a larger group of animals. (2) The results reported here are specific to this acute kidney injury model, which was induced by unilateral occlusion. (3) It would be beneficial to evaluate other time points of acute injury, such as 24 h, for example.

## 5. Conclusions

In conclusion, our data show how the renal inflammatory process induced by I/R triggered a recruitment of both inflammatory mediators and immune cell populations in the kidney and heart tissue. Fundamentally, T cells, B cells, and M1 macrophages were modulated in different proportions in each of the tissues studied, having a greater impact on the kidney compared to the heart, as the former is the focus of injury in the CRS 3 framework. Figure 7 represents a simple summary of the work.

## Figures and Tables

**Figure 1 cells-12-00605-f001:**
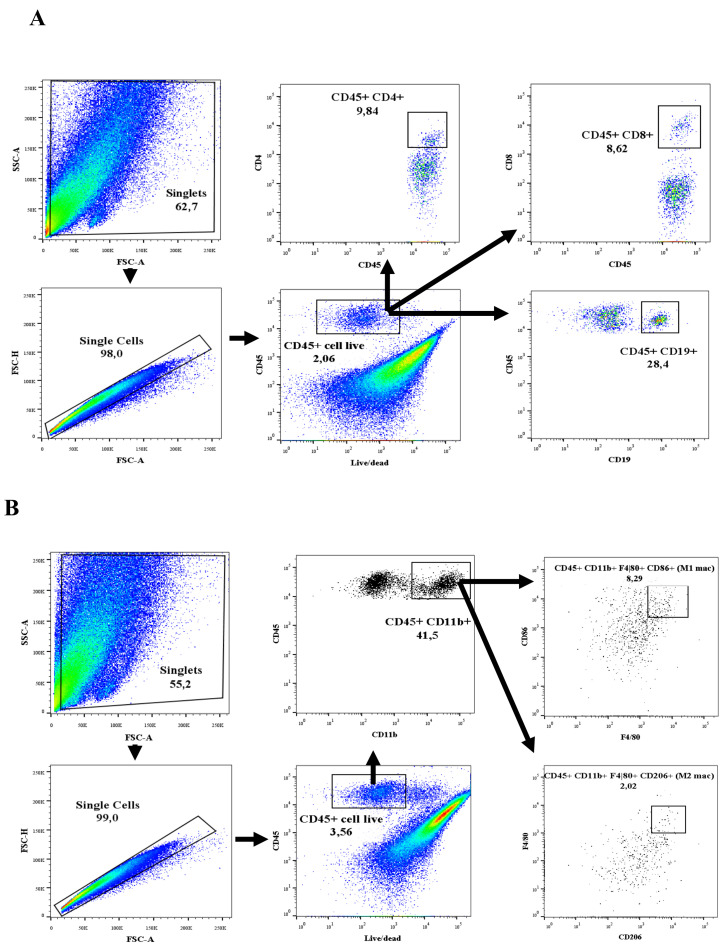
Gating strategy in both kidney and heart to identify populations of (**A**) leukocytes (live CD45+) and lymphocytes (CD45 + CD4+, CD45 + CD8+ and CD45 + CD19+) and (**B**) M1 and M2 macrophages. The data were extracted using the FlowJo program V10.

**Figure 2 cells-12-00605-f002:**
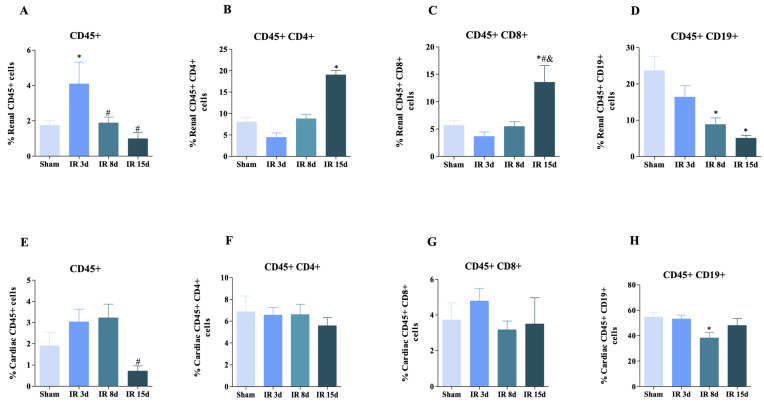
T and B lymphocyte populations are increased in renal and cardiac tissue after renal I/R. Frequency of CD45+, CD4, CD8 e CD19 in the kidney (**A**–**D**) and in the cardiac tissue (**E**–**H**): (**A**) live CD45+ [Sham: n = 9, IR 3d: n = 7, IR 8 days: n = 9, IR 15 days: n = 6]; (**B**) live CD45+ CD4+ [Sham: n = 9, IR 3d: n = 7, IR 8d: n = 9, IR 15d: n = 5]; (**C**) live CD45+ CD8+ [Sham: n = 9, IR 3d: n = 7, IR 8d: n = 9, IR 15d: n = 6]; (**D**) live CD45+ CD19+ [Sham: n = 10, IR 3d: n = 7, IR 8d: n = 9, IR 15d: n = 5] in the left kidney; (**E**) live CD45+ [Sham: n = 9, IR 3d: n = 5, IR 8d: n = 9, IR 15d: n = 6]; (**F**) live CD45+ CD4+ [Sham: n = 9, IR 3d: n = 5, IR 8d: n = 9, IR 15d: n = 4]; (**G**) live CD45+ CD8+ [Sham: n = 10, IR 3d: n = 7, IR 8d: n = 9, IR 15d: n = 6]; (**H**) live CD45+ CD19+ [Sham: n = 9, IR 3d: n = 7, IR 8d: n = 10, IR 15d: n = 6] in the heart. Data are expressed as mean ± SEM, and one-way ANOVA followed by Bonferroni’s post-hoc test were performed. * vs. sham *p* < 0.05; # vs. IR 3d *p* < 0.05; & vs. IR 8d *p* < 0.05.

**Figure 3 cells-12-00605-f003:**
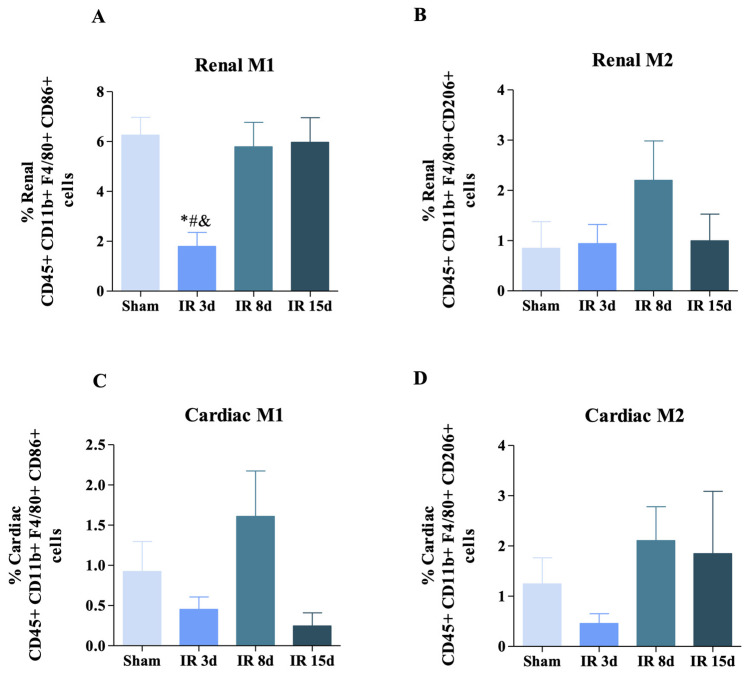
Renal ischemic injury induces macrophage populations in the renal and cardiac tissue. (**A**) CD45+ CD11b+ F4/80+ CD86+ (M1 macrophages) [Sham: n = 10, IR 3d: n = 7, IR 8d: n = 10, IR 15d: n = 6]; (**B**) CD45+ CD11b+ F4/80+ CD206+ (M2 macrophages) [Sham: n = 9, IR 3d: n = 5, IR 8d: n = 9, IR 15d: n = 4] in the left kidney; (**C**) CD45+ CD11b+ F4/80+ CD86+ (M1 macrophages) [Sham: n = 9, IR 3d: n = 7, IR 8d:n = 10, IR 15d: n = 5]; (**D**) CD45+ CD11b+ F4/80+ CD206+ (M2 macrophages) [Sham: n = 9, IR 3d: n = 7, IR 8d: n = 10, IR 15d: n = 4] in the heart. Data are expressed as mean ± standard SEM, and one-way ANOVA followed by Bonferroni’s post-hoc test were performed. * vs. sham *p* < 0.05; # vs. IR 3d *p* < 0.05; & vs. IR 8d *p* < 0.05.

**Figure 4 cells-12-00605-f004:**
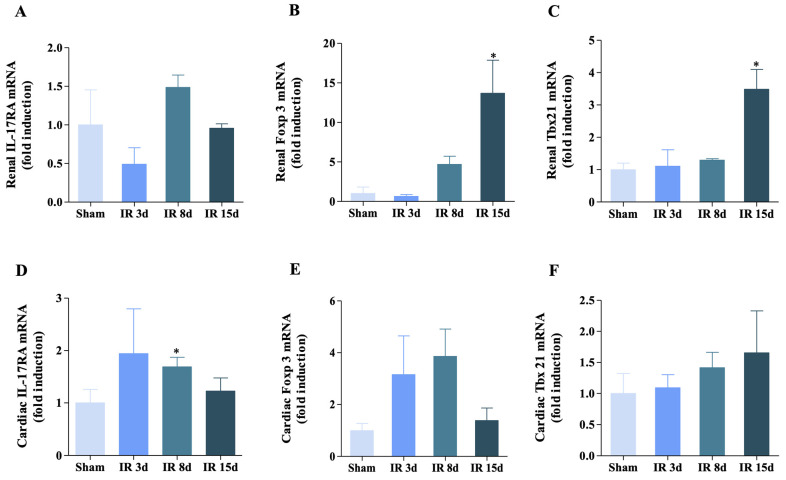
Gene expression of IL-17RA, Foxp3, and Tbx21 was modified for renal I/R in the renal and cardiac tissue. (**A**) IL-17RA mRNA level, (**B**) Foxp3 mRNA level, and (**C**) Tbx21 mRNA level in the left kidney. (**D**) IL-17RA mRNA level, (**E**) Foxp3 mRNA level, and (**F**) Tbx21 mRNA level in the heart. n = 4 for all groups evaluated in both kidney and heart. Data are expressed as mean ± SEM, and one-way ANOVA followed by Bonferroni’s post-hoc test were performed. * vs. sham *p* < 0.05.

**Figure 5 cells-12-00605-f005:**
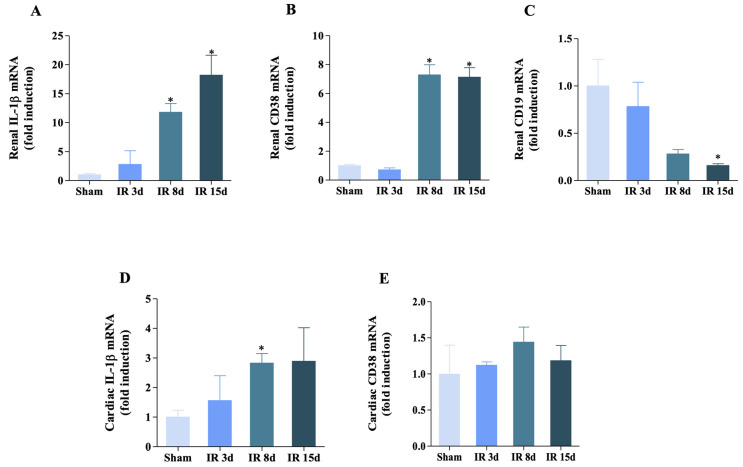
Inflammatory status is modulated in both heart and kidney. (**A**) IL-1β mRNA level, (**B**) CD38 mRNA level, and (**C**) CD19 mRNA level in the left kidney. (**D**) IL-1β mRNA level and (**E**) CD38 mRNA level in the heart. n = 4 for all groups evaluated in both kidney and heart. Data are expressed as mean ± SEM, and one-way ANOVA followed by Bonferroni’s post-hoc test were performed. * vs. sham *p* < 0.05.

**Figure 6 cells-12-00605-f006:**
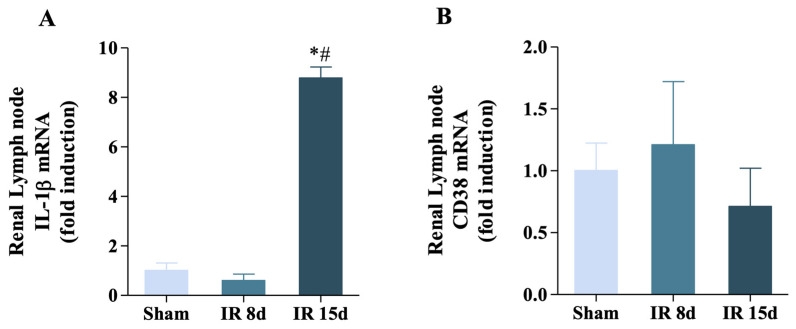
Renal ischemia/reperfusion was able to increase IL-1β expression but not CD38 in the renal lymph node. (**A**) IL-1β mRNA level and (**B**) CD38 mRNA level in the renal lymph node [Sham: n = 5, IR 8d: n = 4, IR 15d: n = 3]. Data are expressed as mean ± SEM, and one-way ANOVA followed by Bonferroni’s post hoc test were performed. * vs. sham *p* < 0.05; # vs. IR 8 days *p* < 0.05.

**Figure 7 cells-12-00605-f007:**
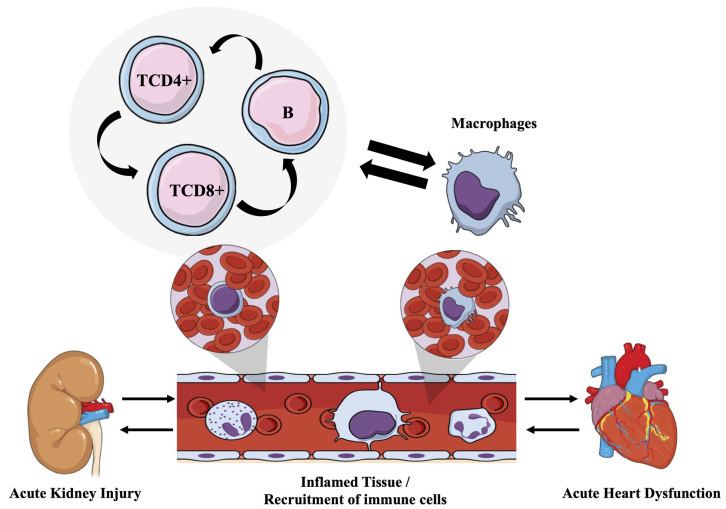
Activation of macrophage and T and B lymphocyte populations induced by renal I/R. I/R induces AKI, producing a local inflammation in the kidney which leads to the recruitment and activation of immune cells such as T and B lymphocyte and macrophages in distinct proportion in the renal tissue. As part of the SCR-3 scenario, the systemic inflammatory process causes alterations in the cardiac tissue where the same cells participate in this injury to the heart but to a lesser degree.

**Table 1 cells-12-00605-t001:** List of primer sequences.

Gene	Forward (5′->3′)	Reverse (5′->3′)
** *Ppia* ** ** *(cyclophilin A)* **	AGCATACAGGTCCTGGCATC	AGCTGTCCACAGTCGGAAAT
** *IL-17ra* **	GTGGCGGTTTTCCTTCAGCCACTTTGTG	GATGCTGTGTGTCCAAGGTCTCCACAGT
** *Foxp3* **	AATAGTTCCTTCCCAGAGTTC	GGGTGGCATAGGTGAAAG
** *Tbx21* **	TGTGTTAATCTCTGACCTGAA	CACCTGAGTCTTCTCTGTT
** *Cd19* **	ACTAGCCTGGACTTCGTTAG	GGTTCTAGGTCGTCAGACTTAT
** *Cd38* **	CGAAGGAGCTTCCAGTAACG	GGTTCTAGGTCGTCAGACTTAT
** *IL-1* ** ** *β* **	AGTTGACGGACCCCAAAAGA	GCTCTTGTTGATGTGCTGCT

## Data Availability

The data is stored following the University’s data management policy and required by FAPESP. The authors declare that all datasets can be provided on demand. The manuscript has no gels or blots.

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
