# Peer review of "Immune Cells Are Differentially Modulated in the Heart and the Kidney during the Development of Cardiorenal Syndrome 3"

_cells, 2023, doi:10.3390/cells12040605_

Round 1

Reviewer 1 Report

The introduction needs to improve with recent references  2021,2022

 The material and method need some details especially in renal I/R

The discussion section needs to improve with recent references 

Author Response

Reviewer #1

  1. The introduction needs to improve with recent references 2021,2022

Answer: We would like to thank the reviewer for this consideration. We've included new references in the new version of the manuscript.

  1. The material and method need some details especially in renal I/R

Answer: We would like to thank the reviewer for the comment. The details of I/R procedure were included in the methodology section, according to Feitoza et al [1]. Briefly, the animals were first weighed and anesthetized with intraperitoneal injection xylazine hydrochloride and ketamine hydrochloride at 100 mg/kg and 200 mg/kg body weight, respectively, diluted in 0.9% saline solution. The renal pedicle was accessed by opening the abdominal cavity, the organs were then plastered in hydrophilic gas. After locating the left renal pedicle of the animal, all the adipose tissue was removed from the region and the left pedicle was isolated with the help of tweezers. The renal pedicle was then occluded by using microvascular clips. After placing the clamp in the renal pedicle, it was possible to observe the immediate change in color of the kidney due to the occlusion of local circulation, indicating the efficiency in inducing renal ischemia. The animals were kept in a thermal blanket for 60 minutes. After this time, the clamps were removed and the viscera were repositioned in the abdominal cavity with the use of flexible rods. The animals first had the peritoneum and then the skin sutured using 6-0 and 3-0 silk thread, respectively. Such a protocol was performed as described in works of the group [2, 3].

The animals were kept warm by indirect lighting until their complete recovery from anesthesia. The groups was divided as follows:

  • Sham Group: underwent the surgical procedure of I/R except the occlusion of the renal pedicle.
  • Group I/R 3d: underwent the surgical procedure of left renal pedicle occlusion for 60 minutes and reperfusion for 3 days.
  • Group I/R 8d: underwent the surgical procedure of occlusion of the left renal pedicle for 60 minutes and reperfusion for 8 days.
  • Group I/R 15d: underwent the surgical procedure of occlusion of the left renal pedicle for 60 minutes and reperfusion for 15 days.

After the surgical procedure, following the established reperfusion times, euthanasia was performed in these three groups, where in each of them one or two mice from the Sham group was included as a control animal, respectively. The thoracic and abdominal cavities of the animals were opened and the left kidney, heart, renal lymph nodes, spleen, were removed for further analysis.

References

  1. Feitoza, C.Q., et al., Inhibition of COX 1 and 2 prior to renal ischemia/reperfusion injury decreases the development of fibrosis. Mol Med, 2008. 14(11-12): p. 724-30.
  2. Trentin-Sonoda, M., et al., Knockout of Toll-Like Receptors 2 and 4 Prevents Renal Ischemia-Reperfusion-Induced Cardiac Hypertrophy in Mice. PLoS One, 2015. 10(10): p. e0139350.
  3. Cirino-Silva, R., et al., Renal ischemia/reperfusion-induced cardiac hypertrophy in mice: Cardiac morphological and morphometric characterization. JRSM Cardiovasc Dis, 2017. 6: p. 2048004016689440.

  1. The discussion section needs to improve with recent references 

Answer: We would like to thank the reviewer for the comment. The discussion section has been updated with recent references and improved.

Reviewer 2 Report

In the work done by Vernier et. al., the authors used a mouse acute kidney injury model to induce cardiorenal syndrome type 3, and studied immune cells distribution and gene expression in heart and kidney tissues. They discovered that immune cells are differentially regulated in those two tissues and the immune system acts as a bridge to connect those two organs. Overall, the manuscript is well written, and easy to understand, all experimental results are clearly presented. However, I find this manuscript is very descriptive, and lack of mechanistic studies, also, the main finding (heart and kidney are connected via immune system) is not new, thus lacking originality and novelty. Other concerns I have:

1.     The authors didn’t say if their animal experiments were approved by the university or institutional animal care committee, this raised my first concern.

2.     As mentioned earlier, all experimental results are clearly presented, but rather descriptive. For each of the main results part, all results were just presented without any interpretation and conclusion, makes them very hard to follow and understand.

Author Response

Reviewer #2

Comments to the Author

In the work done by Vernier et. al., the authors used a mouse acute kidney injury model to induce cardiorenal syndrome type 3, and studied immune cells distribution and gene expression in heart and kidney tissues. They discovered that immune cells are differentially regulated in those two tissues and the immune system acts as a bridge to connect those two organs. Overall, the manuscript is well written, and easy to understand, all experimental results are clearly presented. However, I find this manuscript is very descriptive, and lack of mechanistic studies, also, the main finding (heart and kidney are connected via immune system) is not new, thus lacking originality and novelty. Other concerns I have:

We would like to thank for you your concern regarding the findings and novelty of the study. Indeed, the role of immune cells at different kidney diseases models has been addressed in several research groups, and to a lesser extent, in the heart. Very few studies addressed immune cells in the context of CRS3 by looking at the same time, immune cells in both organs. On top of that, our studies sought to investigate new immune molecules in this context, undoubtedly adding more information for a better understanding and pathophysiology of CRS3.

  1. The authors didn’t say if their animal experiments were approved by the university or institutional animal care committee, this raised my first concern.

Answer: We would like to thank the reviewer for the observation. All procedures and surgical protocols were performed according to the Ethical Principles in Animal Research. We used 57 male C57BL/6 mice, aged 6-8 weeks, weighing between 21-26 g, divided into 4 groups and following the rules of the Ethics Committee on Animal Use of the Federal University of ABC (Protocol CEUA/UFABC No.: 8644151220). Furthermore, during the surgical procedure all the recommendations of the National Council for Control of Animal Experimentation (CONCEA) and the responsible veterinarian of UFABC were followed. We've included these information in the new version of the manuscript.

  1. As mentioned earlier, all experimental results are clearly presented, but rather descriptive. For each of the main results part, all results were just presented without any interpretation and conclusion, makes them very hard to follow and understand.

Answer: We would like to thank the reviewer for the comment. The results section was rewritten to achieve a better compression and interpretation of the data obtained. All the alterations can be observed in the new version of the manuscript.

Reviewer 3 Report

The manuscript entitled, “Immune cells are differentially modulated in heart and kidney during development of cardiorenal syndrome 3" by Vernier et al., aims to evaluate link between Cardiorenal syndrome type 3 during acute kidney injury.  Authors have characterized the macrophage and T and B lymphocyte populations in kidney and heart tissue after acute kidney injury induced by renal I/R. In their model, mice were subjected to a renal I/R protocol by occlusion of the left renal pedicle for 60 min, followed by reperfusion for 3, 8 and 15 days. Methods such as flow cytometry was utilized to characterize immune cell populations, and RT-qPCR was used to evaluate gene expression. Authors found a significant increase of TCD4+, TCD8+ lymphocytes and M1 macrophages to the renal tissue, while B cells in the heart decreased. Their study provides the current evidence linking the AKI generated by renal I/R was able to activate and recruit T and B lymphocytes and macrophages, as well as pro-inflammatory mediators to renal and cardiac tissue.

Overall, this is a well written, significant and well-timed article, this reviewer has certain recommendations that would assist to produce a more comprehensive overview of the topic: 

Comments:

1 Did authors find any difference in TNF-alpha expression in blood, heart, and kidney in their study?

2, The English of manuscript can be polished (minor).

3, The authors should cross-check all abbreviations in the manuscript. Initially, define in full name followed by abbreviation.

4, Authors should write a paragraph about immune cells and their effect on
heart failure (PMID: 16751419, PMID: 36465455, PMID: 36337927; PMID:
34630414; PMID: 34043424; 34119620 etc).

5, Authors can include the limitations to their study. 

6, instead of symbols such as * etc, exact p value can be place in graphs. And in all figure legends also account the statical test performed.

Author Response

Reviewer #3

  1. Comments to the Author

The manuscript entitled, “Immune cells are differentially modulated in heart and kidney during development of cardiorenal syndrome 3" by Vernier et al., aims to evaluate link between Cardiorenal syndrome type 3 during acute kidney injury.  Authors have characterized the macrophage and T and B lymphocyte populations in kidney and heart tissue after acute kidney injury induced by renal I/R. In their model, mice were subjected to a renal I/R protocol by occlusion of the left renal pedicle for 60 min, followed by reperfusion for 3, 8 and 15 days. Methods such as flow cytometry was utilized to characterize immune cell populations, and RT-qPCR was used to evaluate gene expression. Authors found a significant increase of TCD4+, TCD8+ lymphocytes and M1 macrophages to the renal tissue, while B cells in the heart decreased. Their study provides the current evidence linking the AKI generated by renal I/R was able to activate and recruit T and B lymphocytes and macrophages, as well as pro-inflammatory mediators to renal and cardiac tissue.

Overall, this is a well written, significant and well-timed article, this reviewer has certain recommendations that would assist to produce a more comprehensive overview of the topic:

  1. Did authors find any difference in TNF-alpha expression in blood, heart, and kidney in their study?

Answer: We would like to thank the reviewer for the comment. The evaluation of TNF-alpha expression in blood, heart and kidney was not the focus of our study, but there are studies showing that this inflammatory cytokine will present high levels in the setting of CRS-3, including previous study of our lab [4-5]. As part of the tissue damage generated for renal I/R, the immune system is activated secreting pro-inflammatory molecules into the circulation, with potential wide-reaching effect such as TNF-alpha, which through the blood will reach the damaged site, generating and contributing to the exacerbation of inflammation, such as kidney and heart tissue, organs of interest in this study [6-9]. As a consequence of the inflammatory action of TNF-alpha, it is possible to observe mesangial cell death that will lead to pathophysiological changes in the glomerulus [10].

References

  1. Trentin-Sonoda, M, Da Silva, RC, Kmit F, Abrahão, MV, Monnerat Cahli, G, et al, Carneiro-Ramos, MS. Knockout of Toll-Like Receptors 2 and 4 Prevents Renal Ischemia-Reperfusion-Induced Cardiac Hypertrophy in Mice. PlosOne, v. 10, p. e0139350, 2015.
  2. Junho CVC, González-Lafuente L, Neres-Santos RS, Navarro-García, JA Rodríguez-Sánchez E, Ruiz-Hurtado G, Carneiro-Ramos MS. Klotho relieves inflammation and exerts a cardioprotective effect during renal ischemia/reperfusion-induced cardiorenal syndrome. Biomedicine & Pharmacotherapy Volume 153, September 2022, 113515.
  3. Doi, K., Kidney-Heart Interactions in Acute Kidney Injury. Nephron, 2016. 134(3): p. 141-144.
  4. Di Lullo, L., et al., Cardiorenal acute kidney injury: Epidemiology, presentation, causes, pathophysiology and treatment. Int J Cardiol, 2017. 227: p. 143-150.
  5. Ronco, C., A. Bellasi, and L. Di Lullo, Cardiorenal Syndrome: An Overview. Adv Chronic Kidney Dis, 2018. 25(5): p. 382-390.
  6. Wang, J., et al., New insights into the pathophysiological mechanisms underlying cardiorenal syndrome. Aging (Albany NY), 2020. 12(12): p. 12422-12431.
  7. Prastaro, M., et al., Cardiorenal syndrome: Pathophysiology as a key to the therapeutic approach in an under-diagnosed disease. J Clin Ultrasound, 2022. 50(8): p. 1110-1124.

  1. The English of manuscript can be polished (minor).

Answer: We would like to thank the reviewer for the suggestion. The english  was corrected by a Proofreading service English Consulting Brazil (attached).

  1. The authors should cross-check all abbreviations in the manuscript. Initially, define in full name followed by abbreviation.

Answer: We would like to thank the reviewer for the observation. We've checked all the abbreviations.

  1. Authors should write a paragraph about immune cells and their effect on heart failure (PMID: 16751419, PMID: 36465455, PMID: 36337927; PMID:34630414; PMID: 34043424; 34119620 etc).

Answer: We would like to thank the reviewer for the suggestion. We've included a paragraph about immune cells in the introduction section.

  1. Authors can include the limitations to their study.

Answer: We would like to thank the reviewer for the comment. We've included a paragraph about the limitations of our study at the end of discussion.

  1. Instead of symbols such as * etc, exact p value can be place in graphs. And in all figure legends also account the statical test performed.

Answer: We would like to thank the revisor for this consideration. We've included the p value and the statistical test used in each figure. 
